

# Evaluating the performance of five different chemical ionization techniques for detecting gaseous oxygenated organic species

Matthieu Riva[1,2], Pekka Rantala[1], Jordan E. Krechmer[3], Otso Peräkylä[1], Yanjun Zhang[1], Liine Heikkinen[1], Olga Garmash[1], Chao Yan[1], Markku Kulmala[1,4], Douglas Worsnop[1,3], Mikael Ehn[1]

[1] Institute for Atmospheric and Earth System Research / Physics, Faculty of Science, University of Helsinki, Helsinki, 00140, Finland

[2] Univ Lyon, Université Claude Bernard Lyon 1, CNRS, IRCELYON, F-69626, Villeurbanne, France.

[3] Aerodyne Research Inc., Billerica, MA, USA.

[4] Aerosol and Haze Laboratory, Beijing Advanced Innovation Center for Soft Matter Science and Engineering, Beijing University of Chemical Technology (BUCT), Beijing, China

*Correspondence to:*
Matthieu Riva (matthieu.riva@ircelyon.univ-lyon1.fr) & Mikael Ehn (mikael.ehn@helsinki.fi)



## Abstract

The impact of aerosols on climate and air quality remains poorly understood due to multiple factors. One of the current limitations is the incomplete understanding of the contribution of oxygenated products, generated from the gas-phase oxidation of volatile organic compounds (VOC), to aerosol formation. Indeed, atmospheric gaseous chemical processes yield thousands of (highly) oxygenated species, spanning a wide range of chemical formulas, functional groups and, consequently, volatilities. While recent mass spectrometric developments have allowed extensive on-line detection of a myriad of oxygenated organic species, playing a central role in atmospheric chemistry, the detailed quantification and characterization of this diverse group of compounds remains extremely challenging. To address this challenge, we evaluated the capability of current state-of-the-art mass spectrometers equipped with different chemical ionization sources to detect the oxidation products formed from α-pinene ozonolysis under various conditions. Five different mass spectrometers were deployed simultaneously for a chamber study. Two chemical ionization atmospheric pressure interface time-of-flight mass spectrometers (CI-APi-TOF) with nitrate and amine reagent ion chemistries and an iodide chemical ionization time-of-flight mass spectrometer (TOF-CIMS). Additionally, a proton transfer reaction time-of-flight mass spectrometer (PTR-TOF 8000) and a new "Vocus" PTR-TOF were also deployed. In the current study, we compared around 1000 different compounds between each of the five instruments, with the aim to determine which oxygenated VOC (OVOC) the different methods were sensitive to, and identifying regions where two or more instruments were able to detect species with similar molecular formulae. We utilized a large variability in conditions (including different VOC, ozone, $NO_x$ and OH scavenger concentrations) in our newly constructed atmospheric simulation chamber for a comprehensive correlation analysis between all instruments. This analysis, combined with estimated concentrations for identified molecules in each instrument, yielded both expected and surprising results. As anticipated based on earlier studies: the PTR instruments were the only ones able to measure the precursor VOC; the iodide TOF-CIMS efficiently detected many semi-volatile organic compounds (SVOC) with 3 to 5 oxygen atoms; and the nitrate CI-APi-TOF was mainly sensitive to highly-oxygenated organic ($O > 5$) molecules (HOM). In addition, the Vocus showed good agreement with the iodide TOF-CIMS for the SVOC, including also a range of organonitrates. The amine CI-APi-TOF agreed well with the nitrate CI-APi-TOF for HOM dimers. However, the loadings in our experiments caused the amine reagent ion to be considerably depleted, causing non-linear responses for monomers. This



study explores and highlights both benefits and limitations of currently available chemical ionization mass
spectrometry instrumentation for characterizing the wide variety of OVOC in the atmosphere. While
specifically shown for the case of α-pinene ozonolysis, we expect our general findings to be valid also for a
wide range of other VOC-oxidant systems. As discussed in this study, no single instrument configuration can
be deemed better or worse than the others, as the optimal instrument for a particular study ultimately depends
on the specific target of the study.

## 1. Introduction

Atmospheric aerosols, a mixture of solid and liquid particles consisting of organic and inorganic substances
suspended in the air, have significant impact on climate (Albrecht, 1989; Hallquist et al., 2009;
Intergovernmental Panel on Climate Change, 2014; Twomey, 1977). They are also recognized to adversely
impact air quality and human health, representing nowadays the fifth-ranking human health risk factor,
globally (Gakidou et al., 2017). Depending on the regions, organic aerosol on average contributes 20-90% to
the submicron aerosol mass (Jimenez et al., 2009), with secondary organic aerosol (SOA) as the largest source
of atmospheric organic aerosol (Hallquist et al., 2009; Jimenez et al., 2009). SOA is predominantly formed
through the gas-phase oxidation of volatile organic compounds (VOC), producing oxygenated VOC (OVOC),
which can subsequently condense onto pre-existing aerosol particles. Generally, the more oxidized OVOC, the
lower its volatility is, and the greater is the probability of this compound to partition to the particle phase.
Recently, studies have provided new insights on how highly-oxygenated organic molecules (HOM) can form
faster than previously expected, and at high enough yields to make them a major source of condensing, or even
nucleating, compounds  (Ehn et al., 2014; Jokinen et al., 2015; Kirkby et al., 2016; Wang et al., 2018).

The quantitative assessment of the impact of aerosol on climate remains poorly understood due to a

number of factors, including an incomplete understanding of how VOC oxidation processes contribute to new
particle and SOA formation (Glasius and Goldstein, 2016). Indeed, atmospheric oxidation processes can lead
to the formation of thousands of oxidized products from a single precursor (Glasius and Goldstein, 2016;
Goldstein and Galbally, 2007). As a result of these complex oxidation processes, atmospheric organic species
span an extremely wide range of chemical formulas, structures, and physicochemical properties. Volatilities
range from volatile species present only in the gas phase, via low- and semi-volatile organic compounds



(LVOC and SVOC), to extremely low volatility organic compounds (ELVOC) present mainly in particle phase
(Donahue et al., 2012). The chemical complexity of OVOC poses a major challenge in detecting, quantifying,
and characterizing such a large number and wide variety of organic compounds.

Mass spectrometric techniques, which can detect a large range of species simultaneously, are well-

suited to tackle these challenges. This is underlined by the major role of the mass spectrometers in improving
our understanding of the atmospheric chemical composition over the last twenty years (Breitenlechner et al.,
2017; Ehn et al., 2014; Jokinen et al., 2012; Krechmer et al., 2018; Lindinger et al., 1998; Yuan et al., 2017).
Proton transfer reaction (PTR) has been one of the most used low-pressure ionization techniques since the mid-
1990s (Lindinger et al., 1998). Since then, the PTR-MS technique has been greatly improved in terms of
sensitivity, detection limit and mass resolution by introducing the PTR-TOF-MS (Yuan et al., 2017). The latest
version has detection limits as low as $10^7$ molecules cm$^{-3}$. While such techniques can characterize VOC, the
PTR-MS technique has not been able to measure more oxygenated organic species. This is mostly due to losses
of these low volatile compounds in the sampling lines and on the walls of the inlet (caused e.g. by very low
flow rates), as the instrument was designed to primarily measure volatile compounds.

Several different chemical ionization mass spectrometry (CIMS) methods have been developed,

including low-pressure systems like CF$_3$O$^-$-CIMS for specific detection of oxygenated VOC and SVOC
including hydroperoxides (Crounse et al., 2006), acetate-CIMS for selective detection of organic acids
(Bertram et al., 2011), and the iodide adduct ionization CIMS for the detection of wider range of OVOC,
including alcohols, hydroperoxides and peroxy acids (Lee et al., 2014; Riva et al., 2017). These instruments,
based on negative ion chemistry, can detect oxygenated gas phase compounds at concentrations as low as ~$10^6$
molecules cm$^{-3}$. Finally, the discovery of the HOM was possible due to the development of nitrate chemical
ionization source connected to an atmospheric pressure interface time-of-flight mass spectrometer (CI-APi-
TOF) (Ehn et al., 2014; Jokinen et al., 2012).The selectivity and high sensitivity for molecules containing
many functional groups (detection limit below $10^5$ molecules cm$^{-3}$) of the nitrate CI-APi-TOF makes this
instrument perfect for detecting HOM and even certain radicals (e.g., peroxy radicals). As part of the rapid
development in gas-phase mass spectrometry, several new reagent ion chemistries have been tested over the
last few years. With improvements in sensitivity and/or selectivity, new methods are now able to detect a wide



variety of oxygenated species, including radicals and stabilized Criegee intermediates (Berndt et al., 2015,
2017, 2018; Breitenlechner et al., 2017; Hansel et al., 2018; Krechmer et al., 2018).
The selectivity and sensitivity of the different ionization chemistries makes it impossible for one mass
spectrometer to be able to measure the full range of VOC and OVOC present in the atmosphere. Hence, only
a simultaneous deployment of several mass spectrometry techniques can provide a comprehensive chemical
characterization of the gaseous composition. While such a multi-instrument approach maximizes the fraction
of organic species measured (Isaacman-VanWertz et al., 2017, 2018), a number of questions and limitations
can arise in both laboratory and field measurements. For instance, the extent to which instruments (i) can
measure species with identical molecular composition, (ii) can cover the entire range of oxygenated species,
(iii) can provide constant sensitivity across different conditions, have to be determined. Most studies are
typically limited to one, or perhaps two, mass spectrometers, and then it is also important to know which
fraction of the OVOC distribution these instruments are sensitive to. To our knowledge, systematic
comparisons of the most commonly used, or recently developed, gas-phase mass spectrometers are not yet
available. In this work, we compared the suitability of five different chemical ionization methods (including
iodide TOF-CIMS, nitrate and amine CI-APi-TOFs, a PTR-TOF and the newly developed Vocus PTR-TOF)
for the detection of OVOC formed from α-pinene ozonolysis during a comprehensive chamber study with
varying VOC, $O_3$, and $NO_x$ concentrations. We characterized the time evolution of around 1000 compounds
and explored the capability of these instruments to measure OVOC of different oxygenation level within
different compound groups.
## 2. Experimental Section
### 2.1. Chamber experiments
Experiments were performed at the University of Helsinki in a 2-$m^3$ atmospheric simulation Teflon (FEP)
chamber. The "COALA" chamber (named after the project in which it was constructed: **C**omprehensive
molecular characterization of secondary **O**rganic **A**eroso**L** formation in the **A**tmosphere) was operated under
steady-state conditions, meaning that a constant flow of reactants and oxidants were continuously added to the
chamber, while chamber air was sampled by the instruments. Under the conditions used in this study, the
average residence time in the chamber was ~ 30 min, and the majority of conditions were kept constant for 6



to 12 hours before changing to new conditions. These experiments focused on the characterization of the
oxidation products arising from the α-pinene ($C_{10}H_{16}$) ozonolysis. α-pinene was used for the generation of
oxidation products because it is the most abundant monoterpene emitted by the boreal forests and is one of the
most important SOA precursors on a global scale (Jokinen et al., 2015; Kelly et al., 2018).

The experiments were conducted at room temperature (27 ± 2°C) and under dry conditions (RH <

1%). An overview of the measurements, as well as the experimental conditions, are presented Figure 1. α-
pinene was introduced to the chamber from a gas cylinder, and steady-state concentrations of α-pinene were
varied from 20 to 100 ppb. As alkene ozonolysis yields OH radicals (Atkinson et al., 1997), in some
experiments, ~ 1500 ppm of carbon monoxide (CO) was injected to serve as OH scavenger. 10 to 50 ppb of
$O_3$ was generated by injecting purified air through an ozone generator (Dasibi 1008-PC) and monitored over
the process of the campaign using a UV photometric analyzer (Model 49P, Thermo-Environmental). In the
experiments performed in the presence of $NO_x$, 400 nm LED lights were used to generate NO in the chamber
from the photolysis of the injected $NO_2$. The purified air ([$O_3/NO_x$] and [VOC] reduced to less than 1 ppb and
5 ppb, respectively), generated by an air purification system (AADCO, 737 Series, Ohio, USA) running on
compressed air, was used as a bath gas. Temperature, relative humidity (RH) and pressure were monitored by
a Vaisala Humidity and Temperature Probe (INTERCAP® HMP60) and a differential pressure sensor
(Sensirion SDP1000-L025).
2.2. Mass spectrometers
We deployed five chemical ionization schemes to the COALA chamber in order to characterize the chemical
composition of the gas-phase oxidation products for formed from α-pinene ozonolysis. In this section, we
briefly present each instrument, summarized in Table 1. As each mass spectrometer has slightly different
working principles, references to more detailed descriptions are provided. Specific benefits and limitations,
which were not often discussed in earlier studies, are reviewed in section 2.4. Each of the mass spectrometers
were equipped with a mass analyzer manufactured by Tofwerk AG, either an HTOF (mass resolving power
~5000) or Long-TOF (LTOF, mass resolving power ~10000) version.

In the analysis, we focused primarily on the relative behavior of the ions measured by the different

mass spectrometers. An absolute comparison was also performed, but this approach has larger uncertainty, as



the sensitivity towards every molecule is different in each of the mass spectrometers, depending on molecular
size, functionality, proton affinity, polarizability, etc. We attempted a rough estimate of absolute
concentrations for each instrument, despite the fact that with around a thousand of ions analyzed, it is evident
that we make no claim to be accurate for them all. As will be shown, the concentrations of gas-phase VOC and
OVOC vary up to 7 orders of magnitude, and therefore useful information can still be obtained even in cases
where concentration estimates could be off by an order of magnitude. Details about instruments used in this
study as well as calibrations and instrumental limitations are discussed in the following sections.

2.2.1. PTR-TOF

α-pinene concentration was measured in the COALA chamber by a proton transfer reaction time-of-flight mass
spectrometer (PTR-TOF 8000, Ionicon Analytik Gmbh) – later referred to as "PTR-TOF". The technical
details have been described in detail elsewhere (Graus et al., 2010; Jordan et al., 2009). The sample air from
the COALA chamber was drawn to the instrument using 2-m long PTFE tubing (6 mm o.d, 4 mm i.d.) and a
piece of 20-cm capillary PEEK tubing (1.6 mm o.d., 1 mm i.d.), with the sampling flow of 0.8 LPM (liters per
minute). The instrument was operated using a drift tube at the pressure of around 2 mbar and a drift tube at the
temperature of 60 degrees (ºC). Drift tube voltage was kept at 600 V leading to E/N = 145 Td where E is the
electrical field strength and N the gas number density. With these settings, the primary ion isotope ($H_3^{18}O^+$,
21.0221 Th) level stayed at 4500 cps (counts per second), and the mass resolving power of the HTOF mass
analyzer was ~ 4500. Data were recorded using a time resolution of 10 s. The background of the instrument
was measured approximately every day with VOC-free air generated using a custom-made catalytic converter
heated to 350 ºC (Schallhart et al., 2016).

2.2.2. Vocus

The Vocus PTR-TOF (proton transfer reaction time-of-flight mass spectrometer, Tofwerk AG/Aerodyne
Research, Inc.), later referred to as "Vocus", is based on a new PTR-inlet design (i.e., focusing ion-molecule
reactor, FIMR) with sub-ppt detection limits (Krechmer et al., 2018). Sample air was drawn to the instrument
using 1-m long PTFE tubing (6 mm o.d, 4 mm i.d.), with a flow rate of 4.5 LPM. Most of the sample air was
directed to the exhaust while the actual flow to the Vocus was around 0.15 LPM. The instrument was operated
with 1.0 mbar drift tube pressure, the voltages being 350 (axial) and 400 (radial) and E/N = 120 Td. The signal



level of the instrument had some instability during the campaign, thus the primary ion signal ($H_3O^+$, 19.0178
Th) varied from a few hundred to few thousand cps and the isotope of the second water cluster
($H_2^{18}OH_2OH_3O^+$, 57.0432 Th) was around $10^4 - 10^5$ cps. The much lower signal at $H_3O^+$ was due to a high-
pass band filter, that removes most of the ions < 35 Th (Krechmer et al., 2018). The mass resolving power of
the LTOF mass analyzer was 12000-13000 for the whole campaign. Data were recorded using a time resolution
of 10 s. Zero air was produced with a built-in active carbon filter and background was measured hourly except
during December 15 – 17 due the malfunctioning of the zero air pump.

2.2.3. Iodide

Another deployed instrument was a time-of-flight chemical ionization mass spectrometer (TOF-CIMS,
Tofwerk AG/Aerodyne Research, Inc.), equipped with iodide ($I^-$) reagent ion chemistry– later referred to as
"Iodide". While the molecules could be detected as deprotonated species or as adducts with $I^-$, we restricted
the analysis in this work to ions containing only an iodide adduct, which guarantees detection of the parent
organic compounds without substantial fragmentation. Iodide TOF-CIMS has been described previously and
has high sensitivity towards (multifunctional) oxygenated organic compounds (Iyer et al., 2017; Lee et al.,
2014). The instrument was operated at 1 LPM reagent flow rate into the Ion-Molecule Reaction (IMR) chamber
of the instrument. Iodide ions were generated from methyl iodide ($CH_3I$) using a polonium (Po-210) source.
Sample air was drawn to the instrument using 1 m long PTFE tubing (6 mm o.d, 4 mm i.d.) with the flow rate
of 2 LPM. The IMR was temperature controlled at 40˚C and operated at a nominal pressure of 100 mbar. The
instrument, equipped with an HTOF mass analyzer, was configured to measure singularly charged ions from
1 to 1000 Th with a mass resolving power and time resolution of 4000 – 5000 and 10 s, respectively.

2.2.4. Amine/Nitrate

Two chemical ionization atmospheric pressure interface time-of-flight mass spectrometers (CI-APi-TOF,
Tofwerk AG/Aerodyne Research, Inc.) were also deployed (Ehn et al., 2014; Jokinen et al., 2012). The inlet
was designed to minimize wall losses through the use of coaxial sample (10 LPM) and sheath flows (~ 30
LPM), in order to sample (extremely) low-volatile species which are easily lost to the walls. Two types of
ionization schemes were utilized: the promising new amine reagent ion chemistry (Berndt et al., 2017, 2018)
and the more commonly used nitrate chemistry - later referred to as Amine and Nitrate, respectively. The



Amine has been shown to be sensitive towards a very wide range of OVOC, both closed shell species and
peroxy radicals, from molecules with a few oxygen atoms all the way to HOM (Berndt et al., 2018). The
Nitrate, on the other hand, has mainly been used for detection of HOM (Ehn et al., 2014).
Sample air was drawn to the instruments using a common 1 m long PTFE inlet line (19.05 mm o.d,
16 mm i.d.) with the flow rate being ~ 20 LPM (~ 10 LPM for each mass spectrometer). Nitrate ($NO_3^-$) ions
were formed from nitric acid ($HNO_3$) using an X-ray source while protonated butylamine ($C_4H_{12}N^+$) ions were
produced using butylamine with a 7.5 MBq Am-241 source. $NO_3^-$ or $C_4H_{12}N^+$ ions enter the ion reaction zone
together with a clean sheath air flow, concentric with the sample flow, and the two do not mix turbulently. The
ions are then guided into the sample flow by an electrical field. The residence time in the IMR was ~ 200 ms.
The main reagent ions were $NO_3^-$ (*m/z* 62), $HNO_3NO_3^-$ (*m/z* 125) and $(HNO_3)_2NO_3^-$ (*m/z* 188) for the Nitrate
and $C_4H_{12}N^+$ (*m/z* 74) for the Amine. Both instruments were equipped with LTOF mass analyzers providing a
mass resolving power of 9000 – 10000.
2.3. Calibration of the mass spectrometers
In order to estimate absolute concentrations of all detected molecules, each instrument's signals, using an
averaging period of 15 min, were normalized to the reagent ion signals (to eliminate the influence of changes
affecting all signals in the instruments, e.g., due to degrading response of the detector) followed by
multiplication with a scaling factor. The reagent ion quantity used for normalization is described below,
separately for each instrument. Normalized ion count rates are reported as normalized cps, ncps. The scaling
factors were derived differently for each instrument (details provided below). For Iodide, Nitrate and Amine,
the same factor was used for all ions in the spectrum, while for the PTR instruments the factors were different
depending on the type of molecules (e.g., VOC or OVOC). For the PTR instruments and the Iodide, a duty
cycle correction was applied to compensate for mass-dependent transmission due to the orthogonal extraction
of the mass analyzers. Finally, we emphasize that the scaling factors should not be compared between
instruments as a measure of sensitivity, since multiple factors impact these values, including e.g., the specific
normalization approach and the chosen extraction frequency of the mass analyzers.
The PTR-TOF was calibrated twice using a calibration unit consisting of a calibration gas mixture of
16 different VOC (Apel-Riemer Environmental Inc., USA) that was diluted with clean air purified by a
catalytic converter (1.2 LPM of zero air, and 8 sccm of standard gas), producing VOC mixing ratios of around



7 ppb (parts per billion) (Schallhart et al., 2016). Sensitivities were calculated to be 12.31 ncps ppb$^{-1}$, 27.92
ncps ppb$^{-1}$ and 30.51 ncps ppb$^{-1}$ based on the concentration of the monoterpenes, MVK (methyl vinyl ketone)
and m-/o-xylenes, respectively. PTR-TOF signals were normalized using the sum of the first primary ion
isotope at $m/z$ 21.0221 and the first water cluster isotope at $m/z$ 39.0327 (e.g., Schallhart et al., 2016).
According to common practice, the sensitivities above were scaled to correspond to a situation where the total
reagent ion signal equaled $10^6$ cps.
The Vocus was calibrated four times during the campaign using the same calibration gas mixture as
used for the PTR-TOF. There was variability in the sensitivity during the campaign and therefore the
uncertainty in the Vocus results are slightly larger than normal. Sensitivities were highest for acetone,
maximum around 1800 cps ppb$^{-1}$ and around 650 cps ppb$^{-1}$ for monoterpenes. α-pinene concentration was
retrieved using the authentic standard while the concentrations of the OVOC and $C_{10}H_{14}H^+$ were estimated
using the calibration factor of the MVK and sum of m-/o-xylenes, respectively. MVK and m-/o-xylene
sensitivity was around 1700 cps ppb$^{-1}$ and 700 cps ppb$^{-1}$, respectively. Vocus signals were normalized using
the primary ion signal at $m/z$ 19.0178 only, as the water clusters have a negligible effect on the ion chemistry
inside the FIMR (Krechmer et al., 2018). Due to the high-pass filter that removes almost all the signal at $m/z$
19.0178, we do not report the normalized sensitivities (i.e., in ncps ppb$^{-1}$) for the Vocus, in order to avoid direct
comparisons with the PTR-TOF. Instead, the sensitivities above are given without normalization, although a
normalization was used for the final data.
The Iodide was calibrated twice during the campaign (December 15 and 23) by injecting known
amounts of formic acid to the instrument. Due to unknown reasons, the response of the Iodide decayed
throughout the campaign, and therefore only data measured before December 17 was included. While
normalization should compensate for this type of behavior, this particular instrument utilized a time-to-digital
converter (TDC) acquisition card, which meant the primary ion peak was heavily saturated. Lacking any
isotopic signatures for $I^-$, we found that utilizing a region of the rising edge of the $I^-$ signal (126.5-126.65 Th)
provided a reasonable correction to our data. The sensitivity without normalization was 1.0 cps ppt$^{-1}$ for formic
acid, and following the normalization, this sensitivity was applied for all ions throughout the period where
Iodide data was included in the analysis. We acknowledge that this brings with it a large uncertainty, as the



Iodide has sensitivities ranging over a few orders of magnitude depending on the specific molecule (Lee et al.,

2014).

Standards for OVOC compounds measurable by the Nitrate are still lacking, and this instrument was

therefore not directly calibrated during the campaign. However, to be able to roughly estimate concentrations,
a calibration was inferred by assuming that the molar yield of HOM, i.e., molecules with six or more oxygen
atoms, during α-pinene ozonolysis experiments was 5%. Different values have been reported for the HOM
yield in this system, ranging from slightly above to slightly below 5% (Ehn et al., 2014; Jokinen et al., 2014,
2015). Clearly such an approach yields large uncertainties, and we estimated it here to roughly ± 70%. Earlier
work with more direct calibrations reported an uncertainty of ± 50% (Ehn et al., 2014), and the added 20 p.p.
in this work reflects the increased uncertainty in scaling the sensitivity based on expected HOM yields. This
method requires knowledge on the wall loss rate of HOM in the COALA chamber, which was estimated to be
$1/300\ s^{-1}$ in our study. The resulting calibration coefficient was $2 \cdot 10^{10}$ molecules $cm^{-3}\ ncps^{-1}$, which is similar
as in previous studies (Ehn et al., 2014; Jokinen et al., 2012). As for Nitrate, the Amine was also not calibrated
directly, and in order to achieve an estimate of the concentrations measured by this instrument, we scaled the
sensitivity of the Amine to match that of the Nitrate for specific HOM dimers ($C_{19}H_{28/30}O_{12-17}$ and $C_{20}H_{30/32}O_{12-}$
$_{17}$), which were found to correlate very well between the two instruments (as described in more detail in the
Results section). This approach gave a calibration factor of $6 \cdot 10^8$ molecules $cm^{-3}\ ncps^{-1}$, with similar
uncertainty estimates as for the Nitrate. In the CI-APi-TOFs, the calibration factor is generally close to $10^{10}$
molecules $cm^{-3}\ ncps^{-1}$, but as discussed later in section 3.1, the Amine reagent ion was considerably depleted
during the experiments, which led to the relatively low calibration factor. As mentioned earlier, the scaling
factors should not be compared directly between instruments. The lower value for the Amine is a result of the
normalization rather than an indication of higher sensitivity. This reagent ion depletion also means that the
most abundant species were most likely no longer responding linearly to concentration changes, and therefore
their concentrations can be off by an order of magnitude or more.
2.4. Instrumental limitations and considerations
In this section we aim at highlighting some of the limitations involved when characterizing and quantifying
OVOC measured by online mass spectrometers. The list below is not exhaustive, but addresses several issues
that are relevant for the interpretation of our results.



### 2.4.1. Mass resolving power

One major limitation for all of the mass spectrometers described above, is the mass resolving power, ranging from 4000 to 14000. Even though the new generation of LTOF mass analyzers with higher resolving power can enhance the separation of measured ions, it remains challenging to accurately identify and deconvolve the elemental composition of many ions. Indeed, it is common for one CIMS mass spectrum to include more than 1000 different ions. For high-resolution (HR) peak identification and separation, firstly one needs to generate a list of ions, i.e., "peak list". Its construction can be time-consuming even if only based on one single spectrum, and once conditions change, different ions may appear. For measurements lasting weeks or months, it is nearly impossible to assure that all ions are correctly identified and fitted. If the peak list contains too few ions compared to reality, signals from non-fitted ions will assign the adjacent ions with artificially high signals. On the contrary, if too many closely lying ions are included in the peak list, even small errors in the mass axis determination can cause signal to be fitted to specific ions even though their signals are non-existent. In such extreme cases, with closely overlapping ions, traditional HR analysis becomes impossible.

While less selective detection techniques can sound more useful to monitor and characterize OVOC, spectra acquired using such ionization techniques (e.g., PTR, Iodide or Amine), pose a significant challenge for data analysis and may ultimately provide even less useful information. Statistical analysis techniques can be used in order to better constrain the uncertainties associated with peak fitting, as recently proposed (Cubison and Jimenez, 2015; Stark et al., 2015). These previous studies pointed out that the uncertainties related to the peak fitting can become significant if the overlapping peaks are separated by less than a full-width at half-maximum (Cubison and Jimenez, 2015). This is very often the case for CIMS instruments, and the more the ions overlap, the larger the uncertainty is. Peak fitting becomes increasingly problematic as molecular masses increase, since the number of potential ions increases dramatically with mass.

### 2.4.2. Ionization, declustering and fragmentation

The response of a mass spectrometer to a certain compound is to first approximation a result of two factors: the ionization probability of the neutral molecule, and the detection probability of the formed ion. The ionization process is largely controlled by the stability of the products compared to the primary ions, whether a question of adduct formation or (de)protonation processes. Different reagent ion chemistries have been


320 studied computationally in recent years, successfully reproducing several observations (Berndt et al., 2017;

321 Hyttinen et al., 2015, 2018; Iyer et al., 2016). While a neutral molecule can bind to a reagent ion at the collision

322 limit, the adduct can undergo collision-induced dissociation (i.e., declustering) during transport through the

323 interface to the high vacuum in the mass analyzer. Ultimately, the binding strength of the adduct and the energy

324 of the collisions in the mass spectrometer will define the survival probability of the ions. To address this issue,

325 procedures have been proposed, e.g., to probe the response of adducts to different collision energies (Isaacman-

326 VanWertz et al., 2018; Lopez-Hilfiker et al., 2016), providing critical information on the sensitivity of the

327 instrument.

328  Similarly, to declustering, (de)protonated compounds can undergo fragmentation reactions where

329 molecular bonds are broken. For example, the detection of monoterpenes ($C_{10}H_{16}$) using PTR instruments often

330 shows equally large signals at the parent ion ($C_{10}H_{17}^{+}$) and at a fragment ion ($C_6H_9^{+}$). Also, Iodide adducts

331 have been shown to cause molecules to fragment, as in the case of peroxy acids decomposing to carboxylate

332 anions (Lee et al. 2014). Both declustering and fragmentation processes are associated with the optimization

333 of the voltages of each instrument, which is performed by the instrument operator (Breitenlechner et al., 2017;

334 Krechmer et al., 2018; Lopez-Hilfiker et al., 2016). While using voltage scans to probe such processes is

335 possible, and even desirable, performing, interpreting and utilizing the results, across the mass spectrum and

336 across different conditions, remains challenging and has only been utilized in a few studies to date (Isaacman-

337 VanWertz et al., 2018; Lopez-Hilfiker et al., 2016).

338 ### 2.4.3. Quantification

339 For quantification, the instrument sensitivity is generally determined via calibration standards, while a

340 background level was measured by zero air. The challenges involved in these procedures are highly dependent

341 on the type of compounds to be quantified. As an example, we discuss three kinds of molecules with different

342 volatility: VOC, SVOC and ELVOC.

343 (a) VOC: volatile species are relatively easy to quantify since they can be contained in gas bottles or easily

344  evaporated from standard samples in known quantities. Their responses are also fast due to negligible

345  adsorption/evaporation from walls.

346 (b) SVOC: Many semi-volatile organic compounds (SVOC) are commercially available and can be

347  evaporated in known amounts from liquid standards into the gas phase. However, the nature of SVOC



results in both condensed and gas phases for these species, meaning that once clean air is introduced, the
signal of SVOC will often show a gradual decay over minutes, or even hours, due to evaporation of the
'leftover' from surfaces in the inlet lines and the inlet itself (Pagonis et al., 2017). The procedure to
determine the "correct" blank is not trivial, and the blank will look different depending on whether it is
done at the entrance of the instrument or at the sampling inlet, and depending on the duration of the blank
measurement itself. Another related challenge for SVOC quantification is that temperature fluctuations of
a few degrees may cause net evaporation (temperature increasing) or condensation (temperature
decreasing) of SVOC from sampling lines and the inlet.
(c) ELVOC: For ELVOC, finding standard compounds for calibration remains extremely difficult. Most
organic compounds, including hydroperoxide or acid, with such low volatility are likely to decompose
before evaporating. Thus, their quantification is often inferred from other similar compounds. For the
nitrate CI-APi-TOF, sulfuric acid is often used for calibration, by forming it in-situ from $SO_2$ (Kürten et
al., 2012). This is, to some extent, a similar approach as we took for the Nitrate in this work, scaling it to
the estimated HOM yield, as both methods require knowledge of formation rates from the initial precursors
and loss rates of the formed compound of interest. Other studies have used permeation sources of
perfluorinated carboxylic acids, which are semi-volatile, yet found to bind strongly to nitrate ions.
However, while the calibration is complicated, the blank measurements are often not even needed, for
exactly the same reasons. Whatever contaminants might be present in the system, most are irreversibly
lost to instrument surfaces and unable to evaporate into the gas phase due to the extremely low vapor
pressures. Potential oxidation processes occurring inside the mass spectrometer may be an exception, but
to our knowledge, this has not been reported as a large concern for ELVOC.
In addition to the list above, the response of an instrument to specific molecules may vary according to the
conditions at which they were sampled. Temperature (change) was listed as one consideration, and water
vapor, or relative humidity (RH), is another important limitation for several mass spectrometers described
above  (Breitenlechner et al., 2017; Krechmer et al., 2018; Kürten et al., 2012; Lee et al., 2014; Li et al., 2018).
For chemical ionization techniques, the water vapor can either compete with the OVOC ionization, leading to
decrease of the sensitivity, or stabilize the adduct resulting in an increase of the sensitivity. Alternatively, if a





compound forms a very stable complex, it may have an adduct formation efficiency that is independent of
water vapor. If the sensitivity is RH-dependent, calibrations and blanks should optimally be performed at the
same RH as the sampling in order to be representative. This, in turn, may cause considerable practical
challenges for both RH control and calibration and blank cleanliness.
In summary, recent computational and experimental work has shown that many approaches exist for
optimizing the ability of CIMS instruments to quantify OVOC, including different blanks, calibration methods,
voltage scans, etc. However, all these approaches are very rarely utilized in a single study, simply due to the
immense time and effort required, both during the experiments and during the data analysis, where the results
of all steps need to be incorporated. Ultimately, each study needs to prioritize between producing larger
amounts of data (i.e., performing more measurements) with less capability for detailed quantification, or to
produce a smaller amount of data with more accurate quantification.
## 3. Results and discussion
We applied our five CIMS instruments at the COALA chamber over a period of nearly one month, where we
tried to provide different types of atmospherically relevant oxidation conditions for α-pinene. With such a high
variability in conditions, we compared signals between the mass spectrometers more robustly, even though
certain limitations were inevitable. For example, it is often the case that mass spectra will show some signal at
almost every mass, which can be due to multiple reasons, and it is important to separate when the signal is
truly from the sampled air and not from some internal background or contamination. Similarly, one needs to
assess whether the instrument is measuring the majority of the species with the same elemental composition,
or only detecting a small subset of those compounds due to specific selectivity for one isomer. In addition, an
instrument may be able to detect a certain molecule, but the resulting signal remains unreliable. This may be
the case if the sensitivity is extremely low for the molecule, or if the peak is close to a much larger unrelated
signal, which will create large interferences when performing HR fitting. In both cases the signal is likely to
be influenced by different types of noise.
First, we performed correlation analyses in order to identify signals which were physically
meaningful. We conducted the analysis with the whole dataset (a total of ~ 1000 ions in each instrument) rather
than selectively focusing on individual ions. This comprehensive approach utilized more data, but also resulted



in larger uncertainties as not all fitted ions could be validated for all CIMS. From the correlation analysis we
identified when two instruments agree, i.e., observing identical elemental compositions and having a similar
temporal behavior, concerning some group of compounds. From a subsequent absolute comparison, we
estimated which chemical ionization method was likely to be detecting a certain group of compounds more
efficiently, in cases where the correlation between instrument was high, or which CIMS instruments were
detecting a larger number, or the more abundant, isomers at a certain composition, in cases where the
correlation between instrument was low.
3.1. Instrument comparisons: correlations

3.1.1. Low pressure ionization mass spectrometers

Peak fitting was performed utilizing the Igor-based Tofware or Matlab-based Toftools software (Junninen et
al., 2010) for ion mass-to-charge up to ~ 600 Th, depending on the mass spectrometers. To select which ions
to fit (i.e., include in the peak lists), both the exact masses and the isotopic distributions were used as criteria.
A Pearson correlation coefficient $R$, was calculated between molecules having the same elemental composition
measured by the different instruments. As a practical example, the time series of $C_{10}H_{16}O_3$ measured by Vocus
and Iodide are shown in Figure 1C, and the time series correlation for this compound between the two
instruments was $R = 0.85$. For later comparisons we will use $R$ squared, and in this case $R^2 = 0.73$. For Iodide,
the data set covered only the first half of the campaign, but the other instruments covered nearly the whole
period. This includes a wide variety of conditions, with and without $NO_x$, and therefore high correlations are
very suggestive of two instruments measuring the exact same compound(s) at that specific elemental
composition. However, as an increase in α-pinene is likely to increase almost all measured OVOC signals to
some extent, low positive correlations can arise artificially and should not be over-interpreted. Due to the
selectivity and the sensitivity of the ionization methods, all ions were not observed in all the different
instruments, and thus only a certain fraction of the identified compounds can be compared between mass
spectrometer.
Figure 2 shows the correlation analysis for the low-pressure chemical ionization mass spectrometers,
with marker size scaled by $R^2$. In those figures, the abscissa represents the measured $m/z$ of the compounds
and the y-axis their mass defect, which is calculated as the exact mass of the compound minus the mass rounded





to the closest integer (Schobesberger et al., 2013). For example, the mass of $C_{10}H_{16}O_3$ is 184.110 Da and the
mass defect is + 0.110 Da. The contribution of the reagent ions has been removed in the different Figures. A
mass defect diagram helps to separate the molecules into two dimensions and allows some degree of
identification of the plotted markers.

As expected, the PTR-TOF and the Vocus are strongly correlated for compounds with low (0-3)

oxygen number (Figure 2A). Contrariwise, only a few compounds were identified by the PTR-TOF and the
Iodide with a fairly good correlation (i.e., $R^2 > 0.5$). The correlating compounds included small acids such as
formic and acetic acid. As discussed earlier, the inlet of the PTR-TOF is not well designed to sample OVOC
having low volatility, which explained the lack of correlations for larger and more oxidized products between
the PTR-TOF and the Nitrate CI-APi-TOF. The molecules with the lowest correlations ($R^2 < 0.2$) were not
included in the plots, as the intention is to show regions where instruments agree. If an ion is included in a
peak list, it will always be fit, and thereby a value of $R^2 > 0$ is always expected, filling markers throughout the
MD-mass space.

In addition to VOC, the Vocus was able to measure a large range of OVOC ($m/z$ 150-300 Th) as

revealed in Figure 2B, displaying a very good correlation with species identified by the Iodide. Indeed, most
of the identified compounds have $R^2 > 0.7$. As noted earlier, several different experimental conditions were
tested (Figure 1), and these high correlations indicate that both instruments were likely sensitive to the same
compounds. In other words, as good correlation was seen in this mass range for nearly all compositions, the
Iodide and the Vocus did not seem to be strongly impacted by the exact chemical conformation of the organic
compounds. Interestingly no dimers ($m/z > 300$ Th) were observed with the Vocus, which suggests some
potential limitation of the instrument. As a result, very limited correlation was observed between compounds
measured by the Vocus and the Amine or Nitrate CI-APi-TOFs. The two main exceptions being $C_5H_6O_7$
(178.011 Da) and $C_7H_9NO_8$ (235.033 Da). Note that the latter is less clear, as the correlation is nearly identical
between three instruments (Nitrate, Vocus and Iodide). The lack of correlation was not only due to lack of ion
transmission at higher masses in the Vocus, since the instrument was able to detect some ions up to 400 Th,
including $C_{10}H_{30}O_5Si_5H^+$ and $C_{19}H_{29}O_6NH^+$. One possibility was that since the compounds above ~ 300 Th
were likely to contain hydroperoxides, or in the case of dimers, organic peroxides, the ions might may have
fragmented before detection in the Vocus, either during the protonation or due to the strong electric fields in





the Vocus FIMR. In the case of HOM monomers with more than 7 oxygen atoms, an additional limitation
comes from more abundant and closely overlapping ions in the spectra, impacting the accurate fitting of these
ion signals in the Vocus. From our data set, it was not possible to determine the exact cause(s) for this lack of
sensitivity for larger molecules in the Vocus, but it is possible that changes in instrument operating conditions
can extend the range of molecules detectable using the Vocus in future studies.
As shown in Figure 2C, the Iodide was capable of measuring ions with larger masses (i.e., $m/z > 300$
Th) indicating the detection of more complex (e.g., dimers) and oxygenated compounds than the Vocus. This
was the case in spite of the lower flow rate for the Iodide than the Vocus, and thus less optimal for sampling
of low-volatile species (Table 1). The Iodide seemed to have the widest detection range of the mass
spectrometers deployed in this study, showing high correlation with other instruments for organic molecules
from $C_1$ (like formic acid) to $C_{20}$, as long as the molecules had at least two oxygen atoms. This is in line with
earlier findings that the Iodide is sensitive to most species that are polar or have polarizable functional groups
(Iyer et al., 2017; Lee et al., 2014). However, the correlation with the CI-APi-TOFs was still somewhat limited
($R^2 < 0.7$) for HOM monomers and dimers. One reason may have been that these HOM contain peroxy acid
functionalities, which have been shown to undergo reactions in the Iodide TOF-CIMS (Lee et al., 2014). In
this work, we only analyzed the ions containing $I^-$ as these were believed to be the ones where the parent
molecule remained intact. Another reason for lower correlation was the fact that $I^-$ is less selective than other
ionization methods resulting in many overlapping peaks at the same $m/z$ and ambiguous peak fitting (Lee et
al., 2014; Stark et al., 2015, 2017), similar to the case in the Vocus. This means that although the Iodide and/or
the Vocus might be able to charge a specific molecule, and it would not fragment before detection, the ion may
remain unquantifiable due to highly ambiguous peak fitting as a result of multiple overlapping signals.
3.1.2. Atmospheric pressure interface mass spectrometers
Figure 3 shows similar comparisons as in Figure 2, now for the Nitrate (Figure 3A) and the Amine (Figure
3B). Interestingly, these two instruments show excellent correlation ($R^2 > 0.9$) for dimeric products (molecules
within 350-500 Th), but showed mostly low correlations ($R^2 < 0.6$) with other instruments in the monomer
range. The Nitrate had some agreement with the Iodide for certain monomer compounds, but in the HOM
monomer range where the Nitrate generally saw its largest signals ($C_{10}$ molecules with 7 to 11 oxygen atoms;
Ehn et al., 2014), none of the other instruments showed strongly correlating signatures.
Despite showing signals at almost all OVOC, the Amine presented low correlations for all OVOC
except the dimers. In the Amine the reagent ion was greatly depleted due to the relatively high signals (Figure
4), likely leading to a non-linear response for most of the OVOC, apparently with the exception of the HOM
dimers. It may be that the amine reagent ion formed extremely stable clusters with these dimers, and thus any
collision involving these dimers with the reagent ion (regardless of whether already clustered with an OVOC)
in the IMR lead to an amine-dimer cluster. While the Amine showed very low correlation with the other
instruments for most molecules, it has been demonstrated to be an extremely useful detector of both radicals
and closed shell OVOC under very clean, low-loading flow tube experiments (Berndt et al., 2017, 2018). In
other words, it can provide information on a wide variety of OVOC, but to obtain quantitative information, the
amine CI-APi-TOF has to be used in very diluted system (with very clean air) and at low loadings. Determining
more explicitly the limitations requires further studies, but as a rough approximation, the typical CI-APi-TOF
sensitivity of $\sim 10^{10}$ molecules cm$^{-3}$ ncps$^{-1}$ means that when sampling detectable molecules at $10^{10}$ molecules
cm$^{-3}$ ($\sim 0.4$ ppb), these molecules will have ion signals of equal abundance as the reagent ions. Consequently,
once the concentration of measurable molecules exceeds roughly 100 ppt, the CI-APi-TOF may no longer be
an optimal choice. For the Nitrate CI-APi-TOF, which mainly detects HOM with short lifetimes due to their
low volatilities, this has rarely been a limitation, but for less selective reagent ions, like amines, this can be an
important consideration.
3.2. Instrument comparisons: concentration estimates
Concentrations of the identified compounds were estimated for all the different instruments, as described in
section 2.6. It should be noted that no separate inlet loss corrections were applied. The estimations for the
results of PTR-TOF and the Vocus are the most reliable as both instruments were calibrated using authentic
standards with a proven method, while larger uncertainties in the total measured concentrations are expected
for the Iodide and the CI-APi-TOFs.
With around 1000 identified ions for each instrument, except for the PTR-TOF, we decided to focus
our attention in this section on a few particular compound groups: the most abundant $C_{10}$-monomers (i.e.,
$C_{10}H_{14/16}O_n$), $C_{10}$-organonitrates ($C_{10}H_{15}NO_n$) and dimers ($C_{20}H_{32}O_n$). For the non-nitrate compounds, the
concentrations were measured during a steady-state conditions of December 9 from 15:30 to 23:00 with [$O_3$]
= 25 ppb and [α-pinene] = 100 ppb) during period I (Figure 1 in blue). The organonitrate concentrations were





compared using steady-state conditions of December 20 from 02:45 to 07:45 with $[O_3] = 35$ ppb, $[\alpha\text{-pinene}] =$
100 ppb and NO = 0.5 ppb, during period IV (Figure 1 in purple). Figures 5A-D show the concentrations of
the selected species as a function of oxygen number in the molecules. While we again emphasize that all the
concentrations were only rough estimates, these plots painted a similar picture to the correlation analysis, as
described in more detail in the next paragraphs.

Focusing first on the non-nitrate monomers (Figures 5A-B), for compounds with zero or one oxygen

atoms, the PTR-TOF agreed well with the concentration estimated by the Vocus, while molecules with more
than two oxygen atoms were already close to, or below, the noise level of the PTR-TOF. In contrast, as the
number of oxygen atoms in the molecule reached two or more, the Iodide signal increased, and for most
compounds showed concentrations similar to the Vocus. These two instruments agreed on concentration
estimates fairly well all the way up to an oxygen content of around 9 oxygen atoms, where the measured signals
were close to the instruments' noise levels. However, when comparing to the Nitrate, which is assumed to have
good sensitivity for HOM with 7 or more oxygen atoms, the concentrations suggested by the Vocus and Iodide
for the $O_7$ and $O_8$ monomers were very high. We preliminarily attributed this to an over-estimation of the
concentrations of HOM by these two instruments, possibly due to higher sensitivities towards these molecules
as compared to the compounds used for calibration (i.e., MVK). We also did not correct for potential
backgrounds using the blanks for the Iodide, although measured, since the variability in the blank
concentrations (see also discussion in section 2.4) was large enough to cause artificially high fluctuations in
the final signals. Therefore, we opted to not include such a correction, but also note that even if half the signal
at a given ion was attributable to background in the Iodide, then it would only have a small impact on the
logarithmic scales used in Figure 5. Other possible reasons for this discrepancy was that the Iodide and Vocus
were able to detect isomers that the Nitrate was not, or that the Nitrate sensitivity was under-estimated.
However, considering that the Nitrate HOM signal was scaled to match a 5% molar HOM yield, it was unlikely
that the HOM concentrations can be considerably higher than this. Other estimated parameters involved in the
formation and loss rates of HOM also had uncertainties, but we did not expect any of them to be off by more
than 50%. This concentration discrepancy thus remained unresolved, and will require more dedicated future
studies.



Finally, the quantities estimated using the Amine are significantly lower (1-2 orders of magnitude) for
all monomers when comparing to the other instruments. This was presumably related to the titration of the
reagent ion, which meant that the majority of charged OVOC will undergo multiple subsequent collisions with
other OVOC, potentially losing their charge in the process. The Nitrate had, as expected, very low sensitivity
towards less oxygenated compounds, and its highest detection efficiency for HOM (i.e., molecules with at least
6 oxygen atoms).
The organonitrate comparison in Figure 5C suggested that both the Vocus and the Iodide were efficient
at detecting these compounds, as both instruments agreed well ($R^2 > 0.7$) for $C_{10}$-organonitrates with 5 to 10
oxygen atoms. While organonitrates have been detected before using the Iodide (Lee et al., 2016), this was the
first observation that also the Vocus can detect such compounds efficiently. For larger oxygen content, the
Nitrate again seemed to be most sensitive, showing clear signals above 10 oxygen atoms, where the previous
instruments were already close to noise levels. The Amine seemed worse at detecting organonitrates compared
to non-nitrate monomers.
Neither of the PTR instruments were able to detect any dimers in this study. The Amine and the Nitrate
were able to quantify the widest range of HOM dimers, while the Iodide was able to detect less oxidized dimers
(Figure 5D). Based on the concentration estimates, the Amine's detection range also extended to less oxidized
dimers than the Nitrate, as has already been shown by Berndt et al. (2018). Dimers measured by the Iodide
were more abundant than the ones detected by the Amine, but already from the monomer comparisons we
speculated that the Amine might be underestimating concentrations while the Iodide might be overestimating
them. With the data available to us, we can only speculate on the relative sensitivities of the instruments able
to detect dimers, especially with the Vocus providing no support to the comparison.
One aspect lending credibility to the Amine dimer data, in addition to the good time series correlation
with the Nitrate, was the odd-even oxygen atom patterns visible both in the Amine and Nitrate data. Such a
pattern is to be expected, since the 32 hydrogen atoms in the selected dimers indicates that they have been
formed from $RO_2$ radicals where one had 15 hydrogen atoms (which is what ozonolysis will yield, following
OH loss) (Docherty et al., 2005; Lee et al., 2006; Ziemann and Atkinson, 2012), while the second $RO_2$ had 17
hydrogen atoms (which is the number expected from OH oxidation of an alkene where OH adds to the double
bond). The first $RO_2$ from ozonolysis had 4 oxygen atoms, and further autoxidation will keep an even number



of oxygen atoms, while the opposite was true for the OH-derived $RO_2$ which started from 3 oxygen atoms. In
other words, the major dimers from this pathway should contain an odd number of oxygen atoms after
combination. In the case of $C_{20}H_{30}O_n$ dimers, mainly formed from two ozonolysis $RO_2$, the pattern was
expected to show peaks at even n, which is also the case (not shown).

Such odd-even patterns for the oxygen content was not visible in the Iodide, but the reason remained

unknown. It was possible that the dimers detected by the Iodide might be formed via other pathways, where
such a selectivity did not occur. This topic should be explored further in future studies, since dimers formed
form the oxidation of biogenic compounds are important for new-particle formation, and it is therefore critical
to accurately identify and quantify the formation and evolution of different types of dimers. To date, both
dimers measured by Iodide (Mohr et al., 2017) and Nitrate (Tröstl et al., 2016) have been found to be important
for particle formation from monoterpenes.

### 3.3. Performance in detecting oxygenated species

Figure 6 summarizes our results and depicts the performance of each mass spectrometer in detecting monomer
and dimer monoterpene oxidation products. Molecules of $C_{10}H_{16}O_n$, $C_{10}H_{15}NO_n$ and $C_{20}H_{30}O_n$ were provided
as examples. We emphasized that the oxygen content alone was not the determining factor for whether a certain
type of mass spectrometer will detect a compound, but we utilized this simplified representation in order to
provide an overview of the performances of the different chemical ionization schemes. The results were
primarily based on the correlation analysis from section 3.1, and as apparent from the y-axis, this comparison
was only qualitative. However, our aim was to provide an easy-to-interpret starting point, especially for new
CIMS users wanting to compare different available techniques.

For monomer compounds without N-atoms, shown in Figure 6A, the PTR-TOF was limited to the

detection of VOC, while the Vocus was additionally able to measure a large range of OVOC, up to at least 5-
6 oxygen atoms. The Iodide detected OVOC with oxygen content starting from ~3 atoms, but did not seem to
efficiently observe HOM monomers (i.e., $C_{10}H_xO_{>7}$). While being a very promising instrument for a broad
detection of OVOC, the performance of the Amine was limited in our study due to a significant drop of the
reagent ion to ~ 40% of the total signal. Therefore, the Amine was marked with a shaded region rather than a
line, with the lower limit based roughly on its usefulness under the conditions we probed, while the upper limit
was an estimate based on findings in a cleaner system with low loadings (Berndt et al., 2018). Finally, the



Nitrate was mainly selective towards HOM. The detection and quantification of monomeric OVOC containing
5 to 8 oxygen atoms remained as the most uncertain, since there were inconsistencies in both concentration
and correlation between the Nitrate, measuring the more oxygenated species, and the Vocus/Iodide, which
detected the less oxidized compounds.
In Figure 6B, the suitability for the different instruments was plotted for organonitrate monomers. The
Vocus efficiently detected the less oxidized organonitrates, while the Iodide displayed a good sensitivity for
the same compounds, with the exception of the least oxygenated ones. For larger number of oxygens, the
Nitrate again seemed the most suitable method. For dimers (Figure 6C), neither of the PTR techniques showed
any ability to detect these compounds in our study. We did not extend the lines all the way down to n = 0 for
the compounds, as it was still possible that these methods can be able to detect the least oxidized and most
volatile $C_{20}$ compounds, which might not have been present during our experiments. The Iodide showed some
correlation with the Nitrate, but had good signals mainly in the range of dimers with 4 to 8 oxygen atoms. The
Amine and Nitrate correlated well for the most oxidized dimers, suggesting good suitability for dimer detection
of HOM dimers. The Amine concentrations stayed high, with the expected odd-even pattern in oxygen number,
even at lower oxygen content than the Nitrate, and therefore the suitability extended further towards lower O-
atom contents. Again, the shaded area was based on a combination of our findings and those of Berndt et al.

(2018).

As a final test for each instrument, we estimated how much of the reacted carbon (in ppbC) the
different mass spectrometers can explain. As shown in Figure 7, both the Iodide and Vocus seemed to capture
most of the reacted carbon, within uncertainties. The concentration determined using the Vocus was
overestimated, explaining more carbon than was reacted. Out of the largest contributors to the reacted carbon,
pinonaldehyde ($C_{10}H_{16}O_2$) was not efficiently detected by Iodide, but otherwise most of the abundant
molecules were quantified by both Vocus and Iodide. Any carbon lost by condensation to walls or particles
would not have been quantifiable by any of the instruments in this study. While the Nitrate was calibrated with
an assumption that it can measure 5% of the reacted α-pinene, it only detected less than a tenth of that amount.
The reason was that the HOM it can detect were quickly lost to walls (or particles), and thus the gas-phase
concentration was not equivalent to the branching ratio of the VOC oxidation reaction. In fact, and as revealed
by the slow changes in the times series in Figure 7C, most of the carbon ultimately measured by the Nitrate



was semi-volatile, as such compounds accumulated and reached higher concentration in the chamber, unlike
HOM. Thus, while the Nitrate was able to detect a critical group of OVOC from an aerosol formation
perspective, i.e., HOM, for carbon closure studies (Isaacman-VanWertz et al., 2017, 2018) it will be of limited
use. This again highlights the need to first determine the target of a study before deciding which CIMS
technique is the most useful. For the closure comparison in our study, the overestimations emphasized the need
to perform calibration with an extensive set of OVOC, ideally with monoterpene-oxidation products, in order
to better constrain the sensitivity of the products of interest. The study by Isaacman-VanWertz et al. (2018),
as the only study to achieve full carbon closure during chamber oxidation of α-pinene by OH, also successfully
utilized voltage scanning to determine sensitivities of each compound.

## 4. Conclusions

The primary goal of this work was to evaluate the performance of 5 chemical ionization mass spectrometers
(PTR-TOF, Vocus PTR, Iodide TOF-CIMS, Amine CI-APi-TOF and Nitrate CI-APi-TOF) in the
identification and quantification of a wide variety of products formed in the ozonolysis of α-pinene. In addition,
we wanted to estimate the capabilities of the newly developed Vocus PTR in measuring OVOC species. By
comparing the regions of coverage of the instruments across multiple experimental conditions (i.e., in different
$O_3$, VOC, NO and OH radical concentrations) we demonstrated that current instrumentation captures nearly
the entire range of OVOC, spanning from VOC to ELVOC. The PTR-TOF was only able to measure the most
volatile compounds, while the Vocus appeared to be able to measure both VOC and most of the OVOC up to
5 to 6 oxygen atoms. In combination with the Iodide and Nitrate, most of the OVOC range can be measured.
The Iodide showed good overlap with the Vocus for most SVOC with 3 to 5 oxygen atoms, while the Nitrate
detected mainly products with 6 or more oxygen atoms. No dimer species were observed with either of the
PTR instruments, which might be due to wall losses (likely at least for the PTR-TOF) and/or potential
fragmentation in the instruments. The Amine CI-APi-TOF is a promising technique, as shown in earlier
studies, but it likely requires low loadings in order to not titrate the reagent ion, limiting its utility for many
chamber experiments and, potentially, atmospheric observations. The large uncertainties in attempting a
quantification of the wide variety of species measurable with these mass spectrometers underline the urgent
need of developing robust, simple and complete calibration methods in order to obtain a better estimation of



the concentrations. Finally, it is important to underline that the experimental and analytical procedures
performed by the user will ultimately impact the sensitivity, the selectivity, and the interpretability of the
results attainable from each instrument.

Data availability
All data are available by contacting the corresponding authors.

Acknowledgments
This work was supported by the European Research Council (ERC-StG COALA, grant nr 638703). We
gratefully acknowledge Pasi Aalto, Petri Keronen, Frans Korhonen, and Erkki Siivola for technical support.
O.G. thanks Doctoral Programme in Atmospheric Sciences (ATM-DP) at the University of Helsinki for
financial support. O.P. would like to thank the Vilho, Yrjö and Kalle Väisälä Foundation. We thank the
tofTools team for providing tools for mass spectrometry data analysis.

Author contributions
M.R. and M.E. designed the experiments. Instrument deployment, operation, and data analysis were carried
out by: M.R., P.R. J.E.K., O.P., Y.Z., L.H., O.G. C.Y., and M.E.; M.R., P.R., O.P., and M.E., interpreted the
compiled data set. M.R., P. R. and M.E. wrote the paper. All co-authors discussed the results and commented
the manuscript. The authors declare that they have no conflict of interest.



**Table. 1** Overview and characteristics of the mass spectrometers deployed during the campaign at the COALA
chamber.

| Instrument[a] | Ionization[b] | Resolving power[c] | Sampling flow rate (LPM) | T in IMR[d] (°C) | Residence time in IMR | IMR pressure (mbar) | Inlet length (m) |
|---|---|---|---|---|---|---|---|
| PTR-TOF | Proton transfer | 4500 | 0.8 | 60 | 100 µs | 2.0 | 2 |
| Vocus | Proton transfer | 12000 | 4.5 | 30 | 82 µs | 1.0 | 1 |
| Iodide | I$^-$ adduct | 4500 | 2 | 40 | 94 ms | 100 | 1 |
| Amine | $C_4H_{12}N^+$ adduct | 10000 | 10 | Ambient | 200 ms | Ambient | 1 |
| Nitrate | $NO_3^-$ adduct | 9000 | 10 | Ambient | 200 ms | Ambient | 1 |

[a] *The reagent ion is used as a synonym to name the instrument* [b] *Type of ionization method used for each*
*instrument;* [c] *corresponds to the mass resolution of the instruments under the conditions used in this study.* [d]
*IMR = Ion-molecule reaction chamber, i.e. the region where sample molecules are mixed with reagent ions.*
*The IMR has a different design in each of the instruments, except for the Nitrate and Amine, which are*
*identical.*






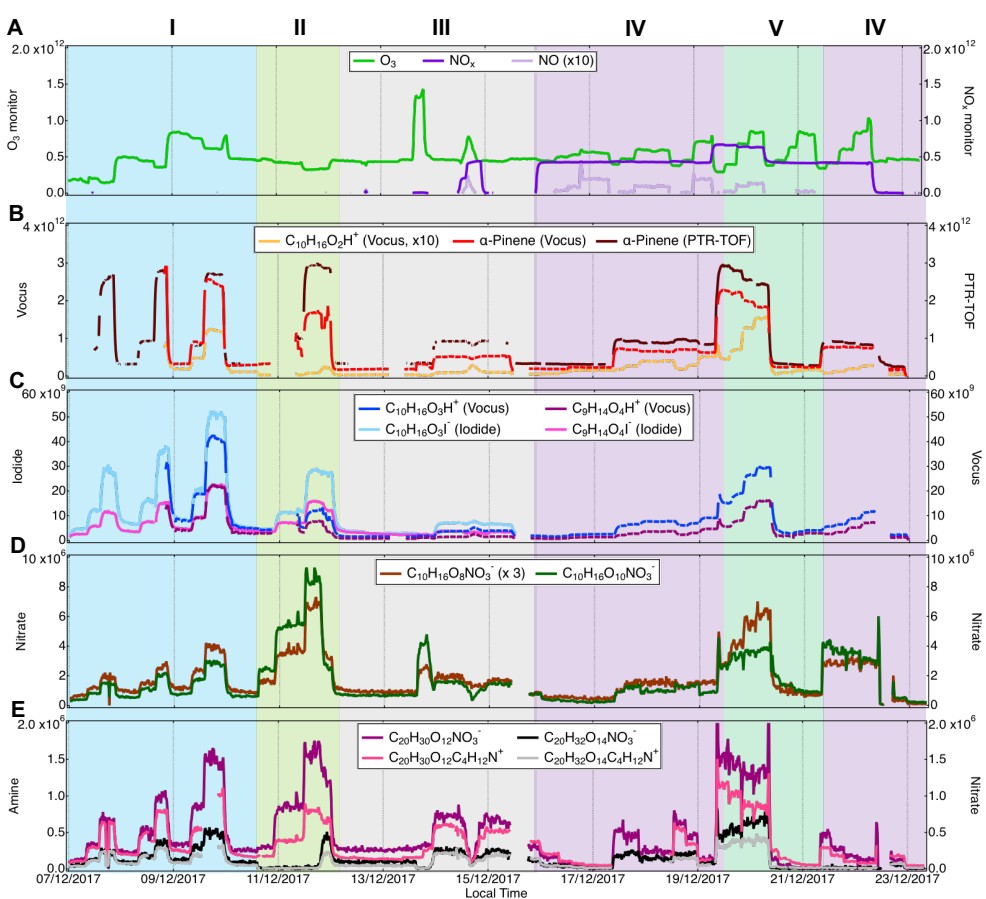


**Figure 1:** Campaign overview, including the concentration of $O_3$, NO, $NO_x$ (A) as well as α-pinene measured

by the Vocus and the PTR-TOF and pinonaldehyde measured using the Vocus (B). Concentrations of pinonic

and pinic acids (Vocus & Iodide) are presented in C, example of HOM monomers from Nitrate (D) and

example of HOM dimers from Amine and Nitrate (E). Concentrations for all the gaseous species are in

molecules $cm^{-3}$, see text for details on quantification. The experiments were separated in 5 types: I: α-pinene

+ $O_3$; II: α-pinene + $O_3$ + CO (as an OH scavenger); III: tests ($NO_2$ injection, $H_2O_2$ injection to generate $HO_2$);

IV: α-pinene + $O_3$ + NO and V: α-pinene + $O_3$ + NO + CO. Concentrations of NO and $C_{10}H_{16}O_8NO_3^-$ are scaled

for clarity.






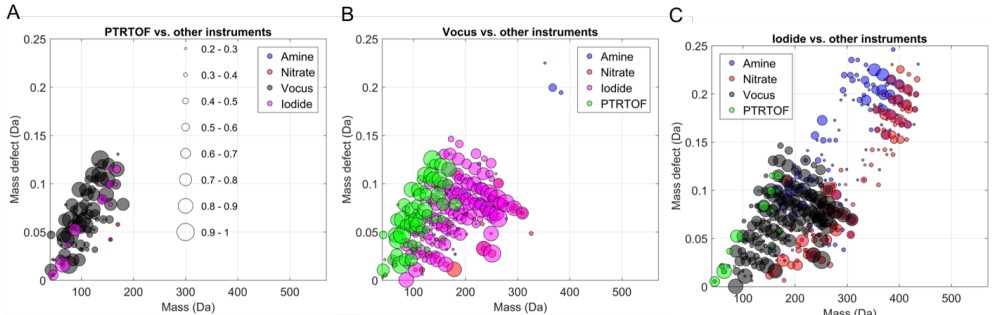


**Figure 2:** Mass defect plots showing the compounds for which time series correlation ($R^2>0.2$) was observed


by the low-pressure chemical ionization mass spectrometers (A) PTR-TOF, (B) Vocus and (C) Iodide. Each


circle represents a distinct molecular composition and the marker area represents the correlation ($R^2$, legend


shown in A) of the time series of that molecule between two different CIMS instruments. The color of each


marker depicts the instrument against which the correlation is calculated.







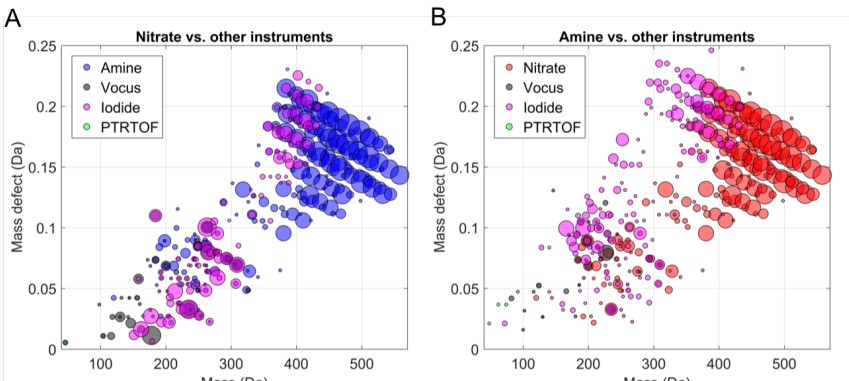


**Figure 3:** Mass defect plots showing the compounds for which time series correlation ($R^2>0.2$) was observed

by the atmospheric-pressure chemical ionization mass spectrometers (A) Nitrate and (B) Amine. Each circle

represents a distinct molecular composition and the marker area represents the correlation ($R^2$, legend shown

in Figure 2A) of the time series of that molecule between two different CIMS instruments. The color of each

marker depicts the instrument against which the correlation is calculated.






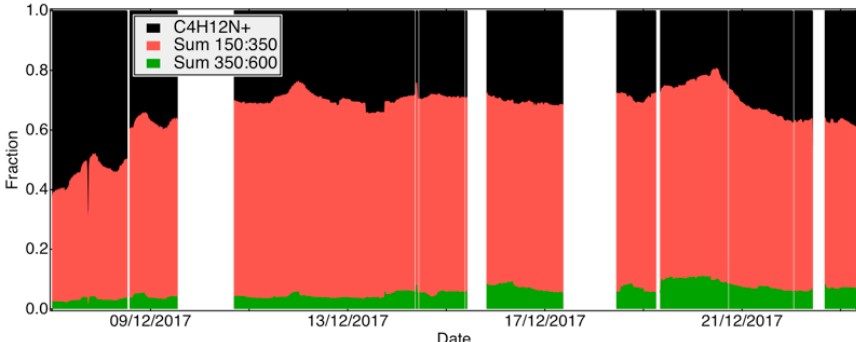


**Figure 4:** Contribution of the reagent ion, sum of ions from m/z 150 to 350 Th and sum of ions from m/z 350

to 600 Th to total ion count throughout the campaign for the Amine CI-APi-TOF. Only a negligible fraction

of the signal was found below $m/z$ 150 Th (excluding $C_4H_{12}N^+$).






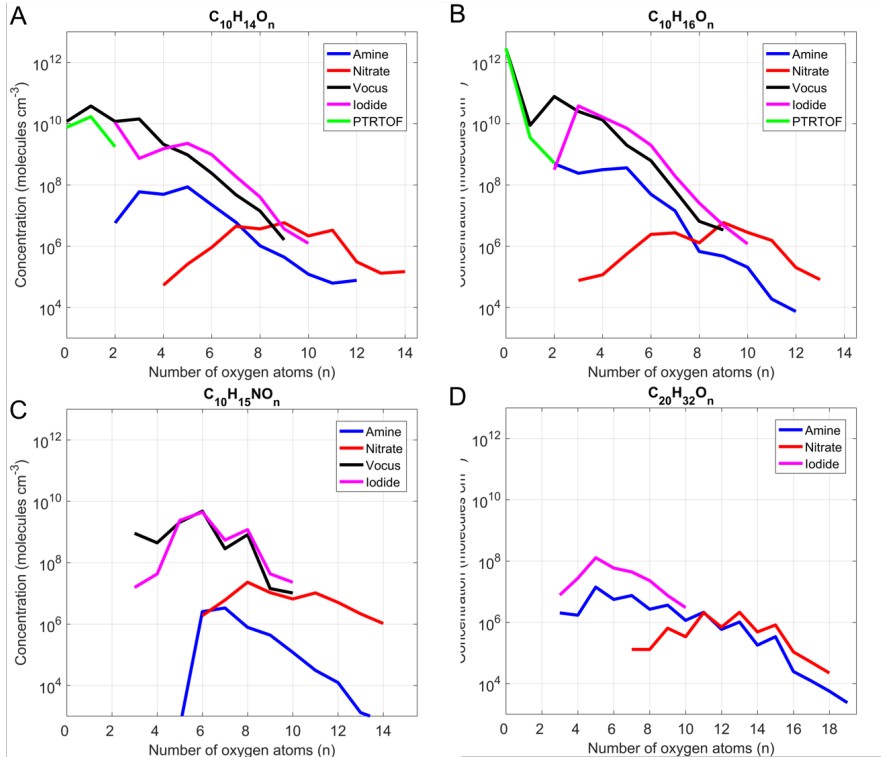


**Figure 5:** Estimated concentrations of the main α-pinene $C_{10}$-monomer oxidation products (A & B), $C_{10}$-

monomer organonitrates (C) and α-pinene dimers (D) by the different mass spectrometers deployed in this

study. The average concentrations were estimated when the system reached steady state in two experiments:

without NO (panels A, B & D), December 9 (15:30-23:00) and with NO (panel C), December 20 (02:45 to

07:45). See text for more details. Data is plotted only for ions for which the average concentrations were higher

than 3 times the standard deviation during the campaign.

715



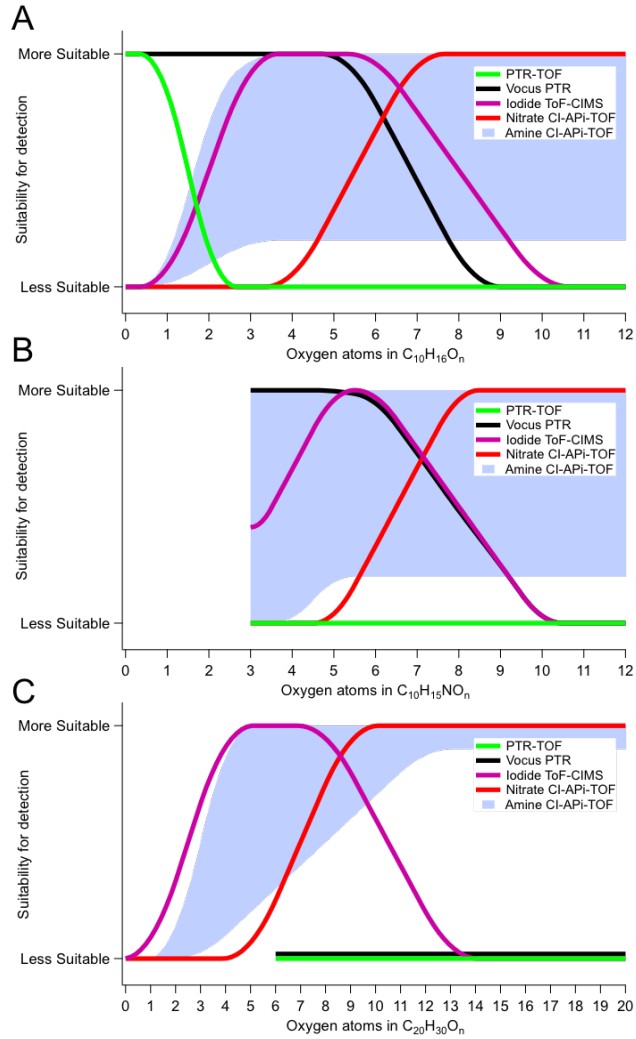

716

**Figure 6:** Estimated detection suitability of the different CIMS techniques for α-pinene and its oxidation

products, plotted as a function of the number of oxygen atoms. Each panel symbolizes a compound group:

monomers (A), organonitrate monomers (B) and dimers (C). The figures are indicative only, as none of the

reagent ion chemistries are direct functions of the oxygen atom content in the molecules. See text for more

details.



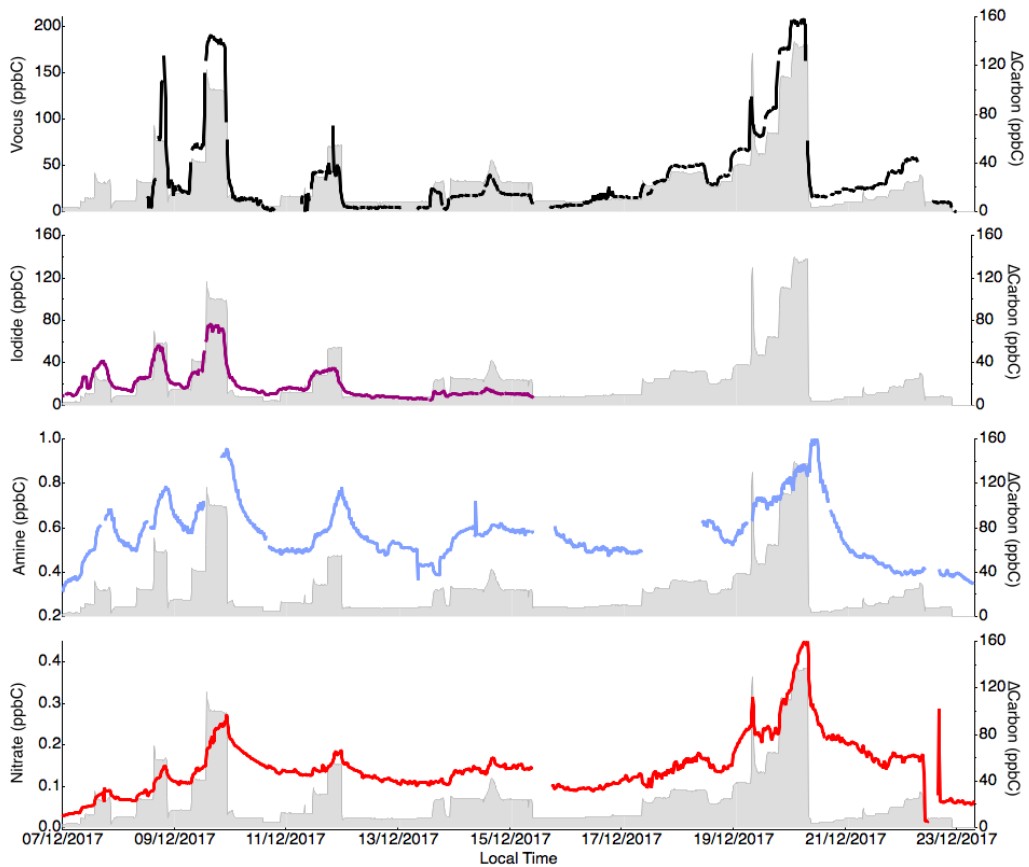

722

**Figure 7:** Concentration ( in ppbC) of the sum of the compounds measured by each instrument (Vocus, Iodide,

Amine and Nitrate) throughout the campaign, compared to the amount of reacted carbon through α-pinene

oxidation. Large uncertainties remain in the quantification of the OVOC for all instruments, but it is clear that

the Iodide and Vocus are able to measure a large fraction of the reacted carbon in the gas phase.




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
