# Peer review of "Evaluating the performance of five different chemical ionization techniques for detecting gaseous oxygenated organic species"

_Atmospheric Measurement Techniques, 2018_

## Referee Comment (RC1) · Anonymous Referee #2 · 10 Jan 2019

General comments

Five different chemical ionization mass spectrometers were applied to measure simultaneously air samples from chamber experiments of $\alpha$–pinene oxidation under varying trace gas concentrations. The authors compare the suitability of the applied CIMS techniques for the detection of different compound classes in the investigated reaction system.

The manuscript is well written and the results are clearly presented. The experimental data are of sufficient quality and the interpretation of the results is expedient. The paper provides valuable new insights in the detection of different compound classes by

five different CIMS techniques. It therefore gives guidance for the selection of CIMS techniques dependent on the scientific problem to be addressed. I recommend its publication considering the following comments.

Specific comments

Chapter 2.3: While for the Nitrate- and with some limitations also for the Amine-CIMS the uncertainty of the concentration estimates are provided, this is lacking for PTRTOFMS, VOCUS and Iodide. At least for the sensitivities mentioned an uncertainty should be provided as this is an essential parameter for an instrument performance, which is stated to be evaluated. The performed calibrations should provide the possibility to derive an acurracy for those instruments.

Page 10, lines 257f: Only data of the Iodide before December 17th was included due to decaying response. What is the reason for addressing data prior to 17th while it becomes invalid afterwards. Has the sensitivity been fallen below a certain threshold or did an abrupt decay of the sensitivity occur? Please specify the reasons for the decision.

Page 11, lines 275f: Please specify what the estimate of a HOM wall loss of 1/300 s-1 is based on.

Page 16, line 407: Please specify the threshold for an 'abundant' signal to be selected for further analysis.

Page 17, line 449: The lack of dimers measured by VOCUS only suggests a potential limitation of the used setup/instrument parameters. It has not been shown that VOCUS is unable to measure dimers under different instrument conditions. → Replace 'instrument' by 'used setup'.

Page 18 , line 462: Referring to the mass to charge ratio for ions measured by Iodide: Does that include the molecular mass of Iodide or has this been removed for comparative reasons?

Chapter 3.1.2: The authors speculate that the amine CIMS is capable of measuring dimers due to the formation of extremely stable amine-dimer-clusters. However this has been observed under conditions where the reagent ion has been depleted significantly due to high trace gas loadings. Therefore the formation of dimers might be positively biased by the excess amount of uncharged monomers forming dimers by clustering with monomer-amine-clusters. Can this possibility being ruled out?

Technical corrections

Page, 11, Line 273 Change '20 p.p' to 20 %

Page,14, Line 362f; Reference for the studies using permeation sources of perfluorinated carboxylic acids is missing

Page 16, Lines 404 – 408: Splitting the sentence into two would improve its readability

Page 16, Line 423: change '. . .methods, all ions were not observed. . .' to '. . .methods, not all ions were observed. . .'

Page 16, Line 425: use plural

Page 23, Line 623: Should be 'Figure 7D'

---

## Referee Comment (RC2) · Anonymous Referee #3 · 14 Feb 2019

General comments: Many previous works discussed the performance of chemical ionization instruments in detecting oxygenated compounds but few of them systematically compared each other. This manuscript comprehensively evaluates the capability of five widely-deployed chemical ionization techniques in the molecule identification, the range of oxygenated species that can detect and the sensitivity under different conditions. The results can be extremely useful for future studies using these mass spectrometry techniques. I recommend the manuscript to be published in AMT after the following comments are considered.

Individual comments:

[Figure]

Line 50-51 "While specifically shown for the case of $\alpha$-pinene ozonolysis, we expect our general findings to be valid also for a wide range of other VOC-oxidant systems." This can be important , but I do not see much discussions on this.

Line 230-232: "For the PTR instruments and Iodide, a duty cycle correction was applied . . . due to the orthogonal extraction of the mass analyzers." Why only for PTR and Iodide CIMS? Why not for nitrate and amine CIMS. Please explain.

Line 244-250 The sensitivity of VOCUS seems to be significantly lower than the reported numbers in Krechmer et al., 2018.

Figure 5: the y-axes in the B and D panels are covered by left ones.

Section 2.3: If the ion-molecule reactions of amines and OVOCs proceed through different mechanisms, more than just adduction reaction, it is better to remind us about it.

Section 2.4 The discussions in this section is very good. Is it possible to move this part to the end of the manuscript, as part of discussions from the findings of this work?

Line 549-550 Organic nitrates usually fragment in PTR-MS instruments (including VO-CUS) by losing the nitrate functional group. The authors should mention this here.

---

## Author Comment (AC1) · 14 Mar 2019

General comments Five different chemical ionization mass spectrometers were applied to measure simultaneously air samples from chamber experiments of $\alpha$–pinene oxidation under varying trace gas concentrations. The authors compare the suitability of the applied CIMS techniques for the detection of different compound classes in the investigated reaction system. The manuscript is well written and the results are clearly presented. The experimental data are of sufficient quality and the interpretation of the results is expedient. The paper provides valuable new insights in the detection

of different compound classes by five different CIMS techniques. It therefore gives guidance for the selection of CIMS techniques dependent on the scientific problem to be addressed. I recommend its publication considering the following comments. We thank the reviewer for careful consideration of our manuscript.

Specific comments Chapter 2.3: While for the Nitrate- and with some limitations also for the Amine- CIMS the uncertainty of the concentration estimates are provided, this is lacking for PTRTOFMS, VOCUS and Iodide. At least for the sensitivities mentioned an uncertainty should be provided as this is an essential parameter for an instrument performance, which is stated to be evaluated. The performed calibrations should provide the possibility to derive an accurracy for those instruments.

The reviewer is correct that more discussion on uncertainties for the different instruments are warranted. However, while it is true that we can derive accuracies for the calibrated compounds, most of the reported concentrations are based on derived sensitivities (e.g., we used the sensitivity of MVK for all oxygenated monoterpene products). For the vast majority of compounds shown in the manuscript, the dominant source of uncertainty comes from the scaling of sensitivity to a certain number that was determined for another compound. Therefore, reporting a single uncertainty for an instrument is problematic.

We have now added details and discussion on the uncertainties into the manuscript to emphasize these points.

Lines 262-267: "The uncertainties for the compounds that were directly calibrated are estimated to be +-20 % for PTR-TOF and Vocus. For other compounds, the uncertainties are much higher due to uncertain ionization efficiencies and potential fragmentation of the compounds with unknown structures. For example, we used sensitivity of MVK for all oxygenated monoterpenes even though all those compounds may have very different fragmentation patterns, transmission rates and/or proton transfer reaction rates from each other. Therefore, we refrain from quantitative estimates of the uncertainties

for these species."

While there would be much more to say, the impact of these uncertainties is not significant for the main conclusions of the manuscript, and therefore we prefer to not go into too much detail on it there. Here, however, we wish to elaborate a bit further.

Take for example the volatile compounds measured by the PTR. These should be the easiest to calibrate out of all the measured compounds, and consequently, also the uncertainty should be the easiest to determine. The accuracy mentioned by the reviewer can be derived from the calibrations. However, other considerations are also important, such as that the standard gas manufacturer (Apel-Riemer) has promised that the reported concentrations in the standard gas mixture are within +-5%. This causes directly 5% uncertainty for the concentrations of the PTR instruments. In addition, mixing the standard gas with the VOC free air (to dilute the VOC concentrations down to a few ppb) relies on the accuracy of the flow measurements themselves. Kajos et al. (2015) also noticed that sensitivities derived from different calibrations - but using exactly the same setup - seemed to be within +-10-20%. This estimate includes all the uncertainties caused by the user (e.g. measuring flows slightly differently) but excludes the possible calibration error of the flow meter and uncertainty of the reported concentrations in the standard gas setup. This estimate may sound surprisingly high but it underlines that calibrating the instruments using even a gaseous standard is not straightforward. Summing all this up, we estimated that the calculated sensitivities of our PTR instruments, for the directly calibrated compounds were within +-20%.

For all the other identified compounds, we needed to use a proxy for the sensitivity as discussed in the manuscript. This approach has, of course, several uncertainties. The proxy does not consider fragmentation - which can be anything from non-existent to almost 100% (e.g. many alcohols such as butanol). In addition, the proton transfer reaction rates which directly affect the sensitivity of the instrument, have large variability (e.g. Zhao et al., 2004), and the reaction rates are poorly known for many oxygenated monoterpene products. Therefore, the proper uncertainty estimates are almost impossible to determine for the derived sensitivities.

Page 10, lines 257f: Only data of the Iodide before December 17th was included due to decaying response. What is the reason for addressing data prior to 17th while it becomes invalid afterward. Has the sensitivity been fallen below a certain threshold or did an abrupt decay of the sensitivity occur? Please specify the reasons for the decision. As mentioned in the paper the sensitivity of the iodide abruptly changed after December 17th, so while it was possible to use the Iodide data during the first half of the campaign it appeared too ambiguous to utilize the data for the rest of our measurements. In addition, we do not know the reasons for this abrupt changed so we decided to exclude these data as were not acquired in a reliable way.

There was indeed an abrupt drop in sensitivity on Dec 17, and therefore we decided to cut the data at this date. This information was now explicitly added to the text.

Lines 269-271: "Due to unknown reasons, the response of the Iodide decayed throughout the campaign, and therefore only data measured before December 17, when a stronger drop occurred, was included for the direct comparison of the non-nitrate OVOC."

Page 11, lines 275f: Please specify what the estimate of a HOM wall loss of 1/300 s-1 is based on.

We now added the following text to the manuscript

Lines 289-292: "This estimate is based on a rough scaling to a slightly smaller chamber (1.5 m3) with active mixing by a fan, where the loss rate was measured to be 0.01 s-1 (Ehn et al., 2014). As our chamber is larger, and our mixing fan was only spinning at a moderate speed, we estimated the loss rates to roughly 3 times lower."

Page 16, line 407: Please specify the threshold for an 'abundant' signal to be selected for further analysis.

The "abundance" related to a comparison between instruments, not to any specific

threshold for further analysis. As the reviewer also mentions in a later comment, this sentence was long and complicated, and thus easily misinterpreted. We chose to remove the second half of the sentence here, as the methodology was described in more detail in a later section.

Page 17, line 449: The lack of dimers measured by VOCUS only suggests a potential limitation of the used setup/instrument parameters. It has not been shown that VOCUS is unable to measure dimers under different instrument conditions. → Replace 'instrument' by 'used setup'.

We agree with the reviewer and we have changed the sentence.

Lines 460-463: "In other words, as good correlation was seen in this mass range for nearly all compositions, the Iodide and the Vocus did not seem to be strongly impacted by the exact chemical conformation of the organic compounds. Interestingly no dimers (mass-to-charge > 300 Th) were observed with the Vocus, which suggests some potential limitation of the instrument or the used settings."

Page 18, line 462: Referring to the mass to charge ratio for ions measured by Iodide: Does that include the molecular mass of Iodide or has this been removed for comparative reasons?

As mentioned on line 430, all the mass to charge ratios used in the Figures and cited in the manuscript are reported without the reagent ion for comparative reasons.

Chapter 3.1.2: The authors speculate that the amine CIMS is capable of measuring dimers due to the formation of extremely stable amine-dimer-clusters. However, this has been observed under conditions where the reagent ion has been depleted significantly due to high trace gas loadings. Therefore the formation of dimers might be positively biased by the excess amount of uncharged monomers forming dimers by clustering with monomer-amine-clusters. Can this possibility being ruled out?

We have looked at the correlation between the most abundant monomers and dimers

measured using the Amine (see Fig.1). As shown in the Figure below no obvious correlation was observed. Therefore, the possible formation of dimers from uncharged monomers clustering with cluster monomers (e.g., C10H16O8 + C10H16O8-C4H12N+ → C20H32O16-C4H12N+) can be ruled out. In addition, the good correlation with the same dimers measured by the Nitrate is extremely unlikely to be a coincidence. Finally, the depletion of reagent ions will not cause an "excess of uncharged molecules", since the fraction of any single neutral molecule becoming charged in the CI inlet is marginal.

Technical corrections Page, 11, Line 273 Change '20 p.p' to 20 % As we are referring to the difference between 50% and 70%, we believe percentage point is the correct unit, and did not make any changes.

Page,14, Line 362f; Reference for the studies using permeation sources of perfluori-nated carboxylic acids is missing Two references have been added (Ehn et al., 2014; Heinritzi et al., 2016).

Page 16, Lines 404 – 408: Splitting the sentence into two would improve its readability As mentioned above, this sentence was amended.

Page 16, Line 423: change '. . .methods, all ions were not observed. . .' to '. . .methods, not all ions were observed. . .' It has been changed

Page 16, Line 425: use plural It has been changed

Page 23, Line 623: Should be 'Figure 7D' The correction has been made as suggested.

References Kajos, M. K., Rantala, P., Hill, M., Hellén, H., Aalto, J., Patokoski, J., Taipale, R., Hoerger, C. C., Reimann, S., Ruuskanen, T. M., Rinne, J., and Petäjä, T.: Ambient measurements of aromatic and oxidized VOCs by PTR-MS and GC-MS: intercomparison between four instruments in a boreal forest in Finland, Atmospheric Measurement Techniques, 8, 4453–4473, 2015.

Zhao, J. and R. Zhang: Proton transfer reaction rate constants between hydronium ion (H3O+) and volatile organic compounds. Atmospheric Environment, 38, 2177–2185,

2004.
* * *
[Figure]

Fig. 1.

---

## Author Comment (AC2) · 14 Mar 2019

General comments: Many previous works discussed the performance of chemical ionization instruments in detecting oxygenated compounds but few of them systematically compared each other. This manuscript comprehensively evaluates the capability of five widely-deployed chemical ionization techniques in the molecule identification, the range of oxygenated species that can detect and the sensitivity under different conditions. The results can be extremely useful for future studies using these mass spectrometry techniques. I recommend the manuscript to be published in AMT after the

following comments are considered.

We thank the reviewer for careful consideration of our manuscript.

Individual comments:

Line 50-51 "While specifically shown for the case of $\alpha$-pinene ozonolysis, we expect our general findings to be valid also for a wide range of other VOC-oxidant systems." This can be important, but I do not see much discussions on this.

We added a paragraph detailing this argument from the abstract. Lines 629-637: "The results in Figure 6 are based on the $\alpha$-pinene ozonolysis system. While we will not speculate too much about the extent to which these findings can be extrapolated to other systems, certain features will remain similar also for other atmospherically relevant reactions. For example, the most oxidized gaseous HOM species will likely have been formed through autoxidation processes, which means that they will contain hydroperoxide functionalities, and could thus be detectable by the Nitrate. Likewise, the HOM, and in particular the dimers, will very likely have low volatilities, requiring high sample flows with minimal wall contact, as in the case of the Eisele-type CI inlets used in the Nitrate and Amine. Several other key features are also expected to be valid in different VOC-oxidant systems, and therefore we believe that our findings are relevant also for many other reaction partners."

Line 230-232: "For the PTR instruments and Iodide, a duty cycle correction was applied ... due to the orthogonal extraction of the mass analyzers." Why only for PTR and Iodide CIMS? Why not for nitrate and amine CIMS. Please explain.

A duty cycle correction is used to compensate for the mass-dependent transmission of the TOF mass spectrometer when performing a calibration. As only the Iodide and the PTR instruments were calibrated using a specific compound (formic acid for the Iodide and monoterpenes for the PTR instruments), we applied this correction for the medium-pressure ionization MS only. For the Amine and the Nitrate the calibration was

inferred using the HOM molar yields, i.e., using the mass range from mass-to-charge 300 to 650 Th. As no duty cycle correction had been done in the studies from which the HOM yields were taken, we decided not to do it either. We have added a sentence to explain this choice. Lines 232-236: "For the PTR instruments and the Iodide, a duty cycle correction was applied to compensate for mass-dependent transmission due to the orthogonal extraction of the mass analyzers. The Amine and Nitrate were calibrated by scaling a wide range of mass-to-charge based on earlier studies, where duty cycle corrections had not been performed. Therefore, we did not apply such a correction for the atmospheric pressure ionization mass spectrometers."

Line 244-250 The sensitivity of VOCUS seems to be significantly lower than the reported numbers in Krechmer et al., 2018. The sensitivity of the Vocus was indeed $\sim$ 10 times lower compared to Krechmer et al. 2018. In this earlier study they used a much larger RF to optimize the sensitivity of the Vocus, which can lead to larger fragmentation. The aim of our measurements was to maximize the amount of compounds detected by each mass spectrometers. So rather than optimizing the sensitivity we made a compromise between fragmentation, sensitivity and mass resolving power.

We have added a sentence to explain this choice Lines 183-184: "The Vocus was operated at a higher water flow than in Krechmer et al. (2018), resulting in a decrease of the OVOC (e.g., HOM) fragmentation but also in a lower sensitivity."

Figure 5: the y-axes in the B and D panels are covered by left ones.

This has been amended.

Section 2.3: If the ion-molecule reactions of amines and OVOCs proceed through different mechanisms, more than just adduction reaction, it is better to remind us about it.

To our knowledge and from previous work (Berndt et al., 2018), the ionization process involving amines predominately forms stable clusters with OVOC. We have added a

sentence to mention this point. Lines 211-215: "The Amine has been shown to be sensitive towards a very wide range of OVOC, both closed shell species and peroxy radicals, from molecules with a few oxygen atoms all the way to HOM (Berndt et al., 2018). Previous work have shown that protonated amines are effective reagent ions, forming stable clusters with OVOC (Berndt et al., 2018)."

Section 2.4 The discussions in this section is very good. Is it possible to move this part to the end of the manuscript, as part of discussions from the findings of this work? We mainly reviewed current limitation. Not sure if it is useful to move this part at the end. Thoughts?

We thank the reviewer for this comment and suggestion, and agree that this is important discussion, which is rarely brought up in the literature. While it would likely receive more attention in the discussion section, we prefer to keep it where it is. Partly because most of the discussion is based on earlier studies, and would therefore greatly dilute the discussion of actual findings from this work. Partly we believe that without this section before the results, readers may have a hard time putting our results into perspective at the time they are presented.

Line 549-550 Organic nitrates usually fragment in PTR-MS instruments (including VO-CUS) by losing the nitrate functional group. The authors should mention this here.

We agree with the reviewer and it is now mentioned in the manuscript. Lines 564-567: "While organonitrates have been detected before using the Iodide (Lee et al., 2016), this was the first observation that also the Vocus can detect such compounds efficiently. However, we cannot exclude that such compounds undergo fragmentation within the drift tube as commonly observed in other PTR instruments (Yuan et al., 2017)."